# XCMAX4: A Robust X Chromosomal Genetic Association Test Accounting for Covariates

**DOI:** 10.3390/genes13050847

**Published:** 2022-05-09

**Authors:** Youpeng Su, Jing Hu, Ping Yin, Hongwei Jiang, Siyi Chen, Mengyi Dai, Ziwei Chen, Peng Wang

**Affiliations:** Department of Epidemiology and Biostatistics, School of Public Health, Tongji Medical College, Huazhong University of Science and Technology, Wuhan 430030, China; yp_s@hust.edu.cn (Y.S.); jing_h2019@163.com (J.H.); pingyin2000@126.com (P.Y.); jhwccc0@sina.com (H.J.); chen_siyi@hust.edu.cn (S.C.); daimengyi0505@163.com (M.D.); happyczw1998@163.com (Z.C.)

**Keywords:** X chromosome, logistic regression, covariates, robust, Graves’ disease

## Abstract

Although the X chromosome accounts for about 5% of the human genes, it is routinely excluded from genome-wide association studies probably due to its unique structure and complex biological patterns. While some statistical methods have been proposed for testing the association between X chromosomal markers and diseases, very a few of them can adjust for covariates. Unfortunately, those methods that can incorporate covariates either need to specify an X chromosome inactivation model or require the permutation procedure to compute the *p* value. In this article, we proposed a novel analytic approach based on logistic regression that allows for covariates and does not need to specify the underlying X chromosome inactivation pattern. Simulation studies showed that our proposed method controls the size well and has robust performance in power across various practical scenarios. We applied the proposed method to analyze Graves’ disease data to show its usefulness in practice.

## 1. Introduction

Many diseases exhibit a gender preference, such as autoimmune diseases, cardiovascular diseases, psychiatric diseases, and cancer, implying that genetic variants on the X chromosome play an important role in sex differences [1,2,3,4,5]. However, most genome-wide association studies (GWAS) routinely exclude the analysis of X-chromosomal variants probably because the X chromosome has a unique structure and complex biological patterns [6,7,8]. Females have one more X chromosome than males, and to balance gene expression on the X chromosome with that of males, one of the female X chromosomes is inactivated in the early embryo [9]. Usually, the process of X chromosome inactivation (XCI) is considered random (XCI-R) [10], i.e., for an X-linked gene, nearly 50% of the cells have the paternal allele active while the rest cells have the maternal allele active. However, studies have shown that skewed XCI (XCI-S) is more biologically plausible [11]. XCI-S is a non-random process, which has been defined as a significant deviation from XCI-R, for instance, the inactivation of one of the alleles in more than 75% of cells [12]. In addition, up to 25% of X-linked genes can escape from XCI (XCI-E) [9]. Both alleles in the genes under XCI-E will be active, which are similar to autosomal genes.

To account for the unique characteristics of the X chromosome, several statistical methods have been developed for testing the association between X chromosomal markers and diseases [13,14,15,16,17,18]. However, very a few of them can adjust for covariates. In large-scale GWAS, spurious associations may occur due to the influence of additional covariates, such as sex, age, and population structure [19,20]. Particularly on the X chromosome, if the sex ratios differ between cases and controls, then sex will be a confounder when the allele frequency of females is unequal to that of males. In practice, a natural way to adjust for covariates is to build a regression model, and logistic regression is generally adopted for binary traits. Based on the logistic regression framework, Gao et al. [15] integrated four tests (${\mathrm{FM}}_{01},{\;\mathrm{FM}}_{02}$, ${\mathrm{FM}}_{F}$, and ${\mathrm{FM}}_{S}$) in the software toolset XWAS. In ${\mathrm{FM}}_{01}$ and ${\mathrm{FM}}_{02}$, three genotypes of females are both coded by 0, 1, and 2, while two genotypes of males are coded by 0 and 1 for ${\mathrm{FM}}_{01}$ and by 0 and 2 for ${\mathrm{FM}}_{02}$. In the latter, males are treated as homozygous females to reflect the dosage compensation relationship between the two sexes. Hence, ${\mathrm{FM}}_{01}$ and ${\mathrm{FM}}_{02}$ assume that the underlying XCI patterns are XCI-E and XCI-R, respectively. On the other hand, ${\mathrm{FM}}_{F}$ and ${\mathrm{FM}}_{S}$ build logistic regressions for females and males separately and then combine the two *p* values using Fisher’s and Stouffer’s methods, respectively. However, these two methods do not take any XCI patterns into consideration and thus may suffer from substantial power loss if the test marker is undergoing XCI. Wang et al. [14] proposed another approach (denoted by maxLR) that can consider four special XCI patterns simultaneously: XCI-R, XCI-E, XCI-S fully toward the normal allele (XCI-SN), and XCI-S fully toward the risk allele (XCI-SR). In their method, three genotypes of females are coded as 0, γ, and 2 under XCI, where $\gamma \in \left(0,2\right)$ measures the degree of skewness of XCI. For instance, $\gamma =0\;\left(2\right)$ represents that all the risk (normal) alleles are inactivated in heterozygous females, which corresponds to the XCI-SN (XCI-SR) pattern. While maxLR has robust performance in power, its *p* value is evaluated based on the permutation procedure, which is very computationally intensive, especially in GWAS. Hence, it is still desirable to develop a robust method that can both adjust for covariates and analytically calculate the *p* value.

To fill this gap, this article proposed a novel statistical method to test the association between X chromosomal markers and a specific disease. Our method, which is also based on logistic regression, is robust because it does not require assigning a specific XCI pattern. Further, our method can compute the *p* value without the resample procedure by directly using the rhombus formula. We implemented an extensive simulation study to compare the performance of our approach with the existing ones. Simulation results showed that our method controls the size well and can maintain relatively high power across a variety of scenarios. Finally, we applied our proposed approach to the Graves’s disease data to demonstrate its practical use.

## 2. Method

Consider an X-linked SNP with deleterious allele *A* and normal allele *a*. Then, there are three possible genotypes for females: *aa*, *Aa*, and *AA*, and two for males: *a* and *A*. We assume a binary variable D for the disease of interest with $D=1\;\left(0\right)$ representing individuals with (without) the disease. $X={\left[x_{1},\cdots ,x_{p}\right]}^{\prime }$ denotes the p covariates that need to be adjusted in the model, where $x_{1}\equiv 1\;$ is the model intercept and $x_{2}$ represents the binary variable with 1 being female and 0 being male. We further assume that the relationship between the phenotype and genotype for individual i can be constructed by the following logistic regression model:(1)$$\log\left(\frac{\mathrm{Pr}\left(D_{i}=1|G_{i},X_{i}\right)}{\mathrm{Pr}\left(D_{i}=0|G_{i},X_{i}\right)}\right)=X_{i}{}^{\prime }\alpha +\beta G_{i}$$
where the subscript i denotes the ith individual, G is the genotypic score, $\alpha ={\left(\alpha_{1},\;\alpha_{2},\cdots ,\alpha_{p}\right)}^{\prime },$ and β represents the regression coefficients for the covariates and the genotypic score. Note that the genotypic score depends on the underlying XCI pattern. According to the coding strategy by Wang et al. [14], $G_{i}$ can be written in the following uniform form
$$G_{i}\left(Z_{1},\;Z_{2}\right)=2I_{i}\left(AA\right)+Z_{1}I_{i}\left(Aa\right)+Z_{2}I_{i}\left(A\right),$$
where $I\left(.\right)\;$ is the indicator function, and $Z_{1}\;$ and $Z_{2}\;$ are unknown parameters depending on the underlying XCI pattern. For instance, when the SNP is undergoing XCI, $Z_{1}\;$ and $Z_{2}$ can be assigned by γ and 2, respectively. In this coding strategy, γ is a measure of the skewness of XCI, and males are treated as homozygous females to reflect the dosage compensation. Table 1 lists the genotypic scores for all five genotypes and the corresponding values of $Z_{1}\;$ and $Z_{2}\;$ under the four special XCI patterns.

We chose the score statistic to test the null hypothesis: β=0 because the association tests for all the SNPs share the same null model. For a total sample size of n, the score function can be derived as
$$U\left(Z_{1,}Z_{2}\right)=\sum_{i=1}^{n}\left\{G_{i}\left(Z_{1},\;Z_{2}\right)\left[D_{i}-\mathrm{Pr}\left(D_{i}=1|X_{i}\right)\right]\right\},$$
where $\mathrm{Pr}\left(D_{i}=1|X_{i}\right)=\frac{exp^{X_{i}{}^{\prime }\hat{\alpha }}}{1+exp^{X_{i}{}^{\prime }\hat{\alpha }}}$ is the disease probability estimated for individual i without considering the genotype (details of the derivation are given in Appendix B). The information matrix of (1) can be written as follows:$$I\left(Z_{1,}Z_{2}\right)=\left(\begin{array}{cc} I_{\beta }\left(Z_{1,}Z_{2}\right) & I_{\beta \alpha }\left(Z_{1,}Z_{2}\right)\\ I_{\beta \alpha }{\left(Z_{1,}Z_{2}\right)}^{\prime } & I_{\alpha } \end{array}\right),$$
where
$$I_{\beta }\left(Z_{1,}Z_{2}\right)=\sum_{i=1}^{n}G_{i}{\left(Z_{1},\;Z_{2}\right)}^{2}\left[1-\mathrm{Pr}\left(D_{i}=1|X_{i}\right)\right]\mathrm{Pr}\left(D_{i}=1|X_{i}\right),$$
$$I_{\beta \alpha }\left(Z_{1,}Z_{2}\right)=\left(\sum_{i=1}^{n}X_{i1}G_{i}\left(Z_{1},\;Z_{2}\right)\left[1-\mathrm{Pr}\left(D_{i}=1|X_{i}\right)\right]\mathrm{Pr}\left(D_{i}=1|X_{i}\right),\cdots ,\sum_{i=1}^{n}X_{ip}G_{i}\left(Z_{1},\;Z_{2}\right)\left[1-\mathrm{Pr}\left(D_{i}=1|X_{i}\right)\right]\mathrm{Pr}\left(D_{i}=1|X_{i}\right)\right),$$
and
$$I_{\alpha }=\left[\begin{array}{ccc} \sum_{i=1}^{n}X_{i1}{}^{2}\left[1-\mathrm{Pr}\left(D_{i}=1|X_{i}\right)\right]\mathrm{Pr}\left(D_{i}=1|X_{i}\right) & \cdots & \sum_{i=1}^{n}X_{i1}X_{ip}\left[1-\mathrm{Pr}\left(D_{i}=1|X_{i}\right)\right]\mathrm{Pr}\left(D_{i}=1|X_{i}\right)\\ \vdots & \ddots & \vdots \\ \sum_{i=1}^{n}X_{ip}X_{i1}\left[1-\mathrm{Pr}\left(D_{i}=1|X_{i}\right)\right]\mathrm{Pr}\left(D_{i}=1|X_{i}\right) & \cdots & \sum_{i=1}^{n}X_{ip}{}^{2}\left[1-\mathrm{Pr}\left(D_{i}=1|X_{i}\right)\right]\mathrm{Pr}\left(D_{i}=1|X_{i}\right) \end{array}\right].$$

Under the null hypothesis: β=0, we have
$$W_{S}=U\left(Z_{1,}Z_{2}\right)V{\left(Z_{1,}Z_{2}\right)}^{-1}U\left(Z_{1,}Z_{2}\right)∼\chi_{1}^{2},$$
where $V\left(Z_{1,}Z_{2}\right)=I_{\beta }\left(Z_{1,}Z_{2}\right)-I_{\beta \alpha }\left(Z_{1,}Z_{2}\right)I_{\alpha }{}^{-1}I_{\beta \alpha }{\left(Z_{1,}Z_{2}\right)}^{\prime }$ is estimated as the variance of $U\left(Z_{1,}Z_{2}\right)$. Therefore, the statistic
$$S\left(Z_{1,}Z_{2}\right)=\frac{U\left(Z_{1,}Z_{2}\right)}{\sqrt{V\left(Z_{1,}Z_{2}\right)}}$$
asymptotically follows a standard normal distribution under the null hypothesis.

Note that the calculation of the test statistic relies on the underlying XCI pattern. Unfortunately, this is generally unknown for a specific SNP. We thereby proposed a robust test referred to as XCMAX4 to account for the four special XCI models. The XCMAX4 statistic is defined as follows:$$\mathrm{XCMAX}4=\max\left(\left|S\left(0,\;2\right)\right|,\left|S\left(1,\;2\right)\right|,\left|S\left(2,\;2\right)\right|,\left|S\left(1,\;1\right)\right|\right)$$

Due to the correlation between the four score tests, XCMAX4 does not follow any classical distributions. We assume that $S\left(0,\;2\right)$, $S\left(1,\;2\right)$, $S\left(2,\;2\right)$, and $S\left(1,\;1\right)$ jointly asymptotically follow a multivariate normal distribution $N\left(0,\;\Sigma \right)$, where 0 is a four-dimensional vector with all elements being 0, and Σ is the correlation matrix with
$$\Sigma =\left[\begin{array}{cc} \begin{array}{cc} 1 & \rho_{\left(0,2\right),\left(1,2\right)}\\ \rho_{\left(1,2\right),\left(0,2\right)} & 1 \end{array} & \begin{array}{cc} \rho_{\left(0,2\right),\left(2,\;2\right)} & \rho_{\left(0,2\right),\left(1,1\right)}\\ \rho_{\left(1,2\right),\left(2,\;2\right)} & \rho_{\left(1,2\right),\left(1,1\right)} \end{array}\\ \begin{array}{cc} \rho_{\left(2,\;2\right),\left(0,2\right)} & \rho_{\left(2,\;2\right),\left(1,2\right)}\\ \rho_{\left(1,1\right),\left(0,2\right)} & \rho_{\left(1,1\right),\left(1,2\right)} \end{array} & \begin{array}{cc} 1 & \rho_{\left(2,\;2\right),\left(1,1\right)}\\ \rho_{\left(1,1\right),\left(2,\;2\right)} & 1 \end{array} \end{array}\right].$$

In the above correlation matrix, $\rho_{\left(z_{11},z_{21}\right),\left(z_{12},z_{22}\right)}$ is the correlation coefficient between $S\left(Z_{11},Z_{21}\right)$ and $S\left(Z_{12},Z_{22}\right)$. Given Σ, we can analytically derive the *p* value of XCMAX4. Particularly, let $f\left(y,0,\;\Sigma \right)$ be the density function of the multivariate normal distribution $N\left(0,\;\Sigma \right);$ then, for a given z>0, the *p* value of XCMAX4 is calculated by
$$P_{r}(\mathrm{XCMAX}4>z)=1-\int \int \int \int_{-z}^{z}f\left(y,0,\Sigma \right)dy.$$

Next, we need to accurately estimate the correlation matrix Σ. To this end, we first build a new model that contains two parameters representing genetic effects as follows:(2)$$\log\left(\frac{\mathrm{Pr}\left(D_{i}=1|G_{i},X_{i}\right)}{\mathrm{Pr}\left(D_{i}=0|G_{i},X_{i}\right)}\right)=X_{i}{}^{\prime }\alpha +\beta_{1}G_{i}\left(Z_{11,}Z_{21}\right)+\beta_{2}G_{i}\left(Z_{12,}Z_{22}\right).$$

The information matrix of (2) can be expressed as follows:$$I\left(Z_{11,}Z_{21},Z_{12,}Z_{22}\right)=\left(\begin{array}{cc} I_{\beta_{1}\beta_{2}}\left(Z_{11,}Z_{21},Z_{12,}Z_{22}\right) & I_{\beta_{1}\beta_{2}\alpha }\left(Z_{11,}Z_{21},Z_{12,}Z_{22}\right)\\ I_{\beta_{1}\beta_{2}\alpha }{\left(Z_{11,}Z_{21},Z_{12,}Z_{22}\right)}^{\prime } & I_{\alpha } \end{array}\right),$$
where
$$\begin{array}{l} I_{\beta_{1}\beta_{2}}\left(Z_{11,}Z_{21},Z_{12,}Z_{22}\right)\\ =\left(\begin{array}{cc} \sum_{i=1}^{n}G_{i}{\left(Z_{11},\;Z_{21}\right)}^{2}\left[1-\mathrm{Pr}\left(D_{i}=1|X_{i}\right)\right]\mathrm{Pr}\left(D_{i}=1|X_{i}\right) & \sum_{i=1}^{n}G_{i}\left(Z_{11},\;Z_{21}\right)G_{i}\left(Z_{12},\;Z_{22}\right)\left[1-\mathrm{Pr}\left(D_{i}=1|X_{i}\right)\right]\mathrm{Pr}\left(D_{i}=1|X_{i}\right)\\ \sum_{i=1}^{n}G_{i}\left(Z_{11},\;Z_{21}\right)G_{i}\left(Z_{12},\;Z_{22}\right)\left[1-\mathrm{Pr}\left(D_{i}=1|X_{i}\right)\right]\mathrm{Pr}\left(D_{i}=1|X_{i}\right) & \sum_{i=1}^{n}G_{i}{\left(Z_{12},\;Z_{22}\right)}^{2}\left[1-\mathrm{Pr}\left(D_{i}=1|X_{i}\right)\right]\mathrm{Pr}\left(D_{i}=1|X_{i}\right) \end{array}\right), \end{array}$$
and
$$I_{\beta \alpha }\left(Z_{11,}Z_{21},Z_{12,}Z_{22}\right)=\left(\begin{array}{ccc} \sum_{i=1}^{n}X_{i1}G_{i}\left(Z_{11,}Z_{21}\right)\left[1-\mathrm{Pr}\left(D_{i}=1|X_{i}\right)\right]\mathrm{Pr}\left(D_{i}=1|X_{i}\right) & \cdots & \sum_{i=1}^{n}X_{ip}G_{i}\left(Z_{11,}Z_{21}\right)\left[1-\mathrm{Pr}\left(D_{i}=1|X_{i}\right)\right]\mathrm{Pr}\left(D_{i}=1|X_{i}\right)\\ \sum_{i=1}^{n}X_{i1}G_{i}\left(Z_{12,}Z_{22}\right)\left[1-\mathrm{Pr}\left(D_{i}=1|X_{i}\right)\right]\mathrm{Pr}\left(D_{i}=1|X_{i}\right) & \cdots & \sum_{i=1}^{n}X_{ip}G_{i}\left(Z_{12,}Z_{22}\right)\left[1-\mathrm{Pr}\left(D_{i}=1|X_{i}\right)\right]\mathrm{Pr}\left(D_{i}=1|X_{i}\right) \end{array}\right)$$

Under the null hypothesis $\beta_{1}=\beta_{2}=0$, the statistic
$$W_{S}=\left[U\left(Z_{11,}Z_{21}\right),U\left(Z_{12,}Z_{22}\right)\right]C{\left(Z_{11,}Z_{21},Z_{12,}Z_{22}\right)}^{-1}{\left[U\left(Z_{11,}Z_{21}\right),U\left(Z_{12,}Z_{22}\right)\right]}^{\prime }$$
asymptotically follows a chi-square distribution with two degrees of freedom, where
$$C\left(Z_{11,}Z_{12},Z_{21,}Z_{22}\right)=I_{\beta_{1}\beta_{2}}\left(Z_{11,}Z_{21},Z_{12,}Z_{22}\right)-I_{\beta_{1}\beta_{2}\alpha }\left(Z_{11,}Z_{21},Z_{12,}Z_{22}\right)I_{\alpha }{}^{-1}I_{\beta_{1}\beta_{2}\alpha }{\left(Z_{11,}Z_{21},Z_{12,}Z_{22}\right)}^{\prime }$$
is the covariance matrix of $U\left(Z_{11,}Z_{12}\right)$ and $U\left(Z_{21,}Z_{22}\right)$. Therefore, the correlation coefficient between $S\left(Z_{11,}Z_{12}\right)$ and $S\left(Z_{21,}Z_{22}\right)$ can be estimated as
$$\frac{\left(1,0\right)C\left(Z_{11,}Z_{21},Z_{12,}Z_{22}\right){\left(0,1\right)}^{\prime }}{\sqrt{\left(1,0\right)C\left(Z_{11,}Z_{21},Z_{12,}Z_{22}\right){\left(1,0\right)}^{\prime }\times \left(0,1\right)C\left(Z_{11,}Z_{21},Z_{12,}Z_{22}\right){\left(0,1\right)}^{\prime }}}$$

Once Σ is estimated, we can calculate the *p* value of XCMAX4. Although the four-dimensional integral can be calculated in the commonly used software (e.g., the mvtnorm package in R, https://cran.r-project.org/web/packages/mvtnorm/index.html, (accessed on 10 April 2022)), the algorithm based on the Quasi-Monte-Carlo procedure needs a lot of computing resources to achieve relatively high accuracy. Hence, it would be still desirable to obtain its analytic form if possible. Fortunately, we can use the rhombus formula [13,21] to obtain the upper bound of the *p*-value of XCMAX4 as follows:$$P\left(\mathrm{XCMAX}4>z\right)\leq 2\left[\Phi \left(z\right)-\Phi \left(-z\right)-1\right]+\frac{4\phi \left(z\right)}{z}\overset{3}{\underset{i=1}{\sum }}\left[\Phi \left(\frac{L_{i\left(i+1\right)}z}{2}\right)+\Phi \left(\frac{\left(\pi -L_{i\left(i+1\right)}\right)z}{2}\right)-1\right],$$
where $\Phi \left(x\right)$ and $\phi \left(x\right)$ denote the cumulative distribution function and probability density function of the standard normal distribution, respectively, and $L_{i\left(i+1\right)}=\arccos\left(\rho_{i\left(i+1\right)}\right)$, where $\rho_{i\left(i+1\right)}$ is the correlation efficient between ith and $\left(i+1\right)$th score statistics. Note the order of four test statistics $S\left(0,2\right),$ $S\left(1,2\right)$, $S\left(2,2\right)$, and $S\left(1,1\right)$ is not specified in the above formula, so 12 kinds of upper bounds can be obtained. Therefore, only the smallest bound among them is adopted as an approximation of the *p* value. As shown in Wang et al. [13], such approximation is very accurate for small *p* values, which would be quite useful in GWAS because the significance level is generally very stringent (e.g., $5\times 10^{-8}$) in such studies.

## 3. Simulation Study

### 3.1. Simulation Settings

We conducted comprehensive simulation studies to compare the performance of XCMAX4 with ${\mathrm{FM}}_{01}$, ${\mathrm{FM}}_{02}$, ${\mathrm{FM}}_{F}$, and ${\mathrm{FM}}_{S}$, all of which can adjust covariates. Note that we did not include the maxLR in our simulations because this method is a permutation-based approach, which would be too time-consuming for GWAS. The data are simulated from the following model:(3)$$\log\left(\frac{\mathrm{Pr}\left(D_{i}=1|G_{i},x_{i2},x_{i3}\right)}{\mathrm{Pr}\left(D_{i}=0|G_{i},x_{i2},x_{i3}\right)}\right)=\alpha_{1}+\alpha_{2}x_{i2}+\alpha_{3}x_{i3}+\beta G_{i},$$
where $x_{2}$ is the binary covariate sex, $x_{3}$ is a continuous covariate, which is sampled from the uniform distribution $U\left(0,1\right),$ and G is the genotype score. The ratio of males to females is assumed to be 1:1 in the general population, so $x_{2}$ follows a binomial distribution $B\left(0.5\right)$. Further, we assume that the genotype of females (*aa*, *Aa*, *AA*) follows a trinomial distribution with probabilities $\left(q_{f0},\;q_{f1},q_{f2}\right)$, while the genotype of males (*a*, *A*) follows a binomial distribution $\left(1-q_{m},\;q_{m}\right)$. Let $q_{f}$ and F be the respective risk allele frequency and the inbreeding coefficient for females. Then, we have $q_{f0}={\left(1-q_{f}\right)}^{2}+Fq_{f}\left(1-q_{f}\right)$, $q_{f1}=2(1-F)q_{f}\left(1-q_{f}\right)$, and $q_{f2}=q_{f}^{2}+Fq_{f}\left(1-q_{f}\right)$. The values of $q_{f}$ and $q_{m}$ are both set to be 0.1, 0.2, and 0.3, so there are nine combinations in total. F is assigned to be 0 and 0.05, where the former implies Hardy–Weinberg equilibrium (HWE) and the latter represents a scenario of Hardy–Weinberg disequilibrium (HWD). The intercept $\;\alpha_{1}$ is fixed at −5. For the coefficients $x_{2}$ and $x_{3}$, we consider two cases for each of them: $\alpha_{2}=\left(0.4005,-0.4005\right)$ and $\alpha_{3}=\left(0.5,\;1.5\right)$. The genetic effect β is set to be 0, 0.1116, 0.15, and 0.1858, where β=0 means no association between the SNP and the disease status, and the other three values of β indicate that the odds ratios of females with genotype *AA* are about 1.25, 1.35, and 1.45. Obviously, the case of β=0 is used to study the size, while the empirical power is investigated in the non-zero β cases.

Note that, when studying the power, we only choose three combinations of $q_{f}$ and $q_{m}$: $\left(0.3,\;0.3\right),\;\left(0.3,\;0.2\right),\;\mathrm{and}\;\left(0.2,\;0.3\right)$ for convenience. The scenarios that the SNP undergoes XCI or escapes from XCI are both considered. For the former, we let γ range from 0 to 2 in increments of 0.5. As such, we have considered various XCI patterns, including XCI-SN, XCI-R, and XCI-SR. Once the XCI pattern is assumed, we can assign the corresponding value for the genotypic score G.

Given the covariates, the genotypic score, and the regression coefficients, we can generate the disease status from the binomial distribution for a large population. Then, we randomly sample 2500 cases and 2500 controls from this population. We find that when $\alpha_{2}=\pm 0.4005$, the proportions of females in cases varied from 40% to 60% in the simulated data. The size is estimated at three nominal levels: $\alpha =1\times 10^{-3},\;1\times 10^{-4},\;\mathrm{and}\;1\times 10^{-5}$ based on 1,000,000 replicates, while the power is only estimated at the nominal level $\alpha =1\times 10^{-4}$ based on 10,000 replicates. The *p* value of XCMAX4 is evaluated by using the rhombus formula.

### 3.2. Results

#### 3.2.1. Size

Table 2 shows the estimated type I error rate at the nominal significance level $\alpha =1\times 10^{-4}$ when HWE holds in the female population. As expected, all the methods controlled the size well in all the scenarios. Although XCMAX4 appears slightly conservative in some scenarios, its *p* values are similar to the nominal level. We also simulated the scenarios of HWD (F=0.05). However, we observed that the performances of all the tests were similar to those of Table 2, and HWD in females had little impact on the size. Therefore, the simulation results with non-zero F are presented in the Appendix A (Appendix A). The results of type I error rates estimated at the nominal level $\alpha =1\times 10^{-3}$ and $\alpha =1\times 10^{-5}$ are also given in Appendix A (Appendix A). As can be seen, XCMAX4 still had the correct size in general, except being slightly conservative at $\alpha =1\times 10^{-3}$.

#### 3.2.2. Power

Figure 1, Figure 2 and Figure 3 plot the powers of XCMAX4, ${\mathrm{FM}}_{01}$, ${\mathrm{FM}}_{02}$, ${\mathrm{FM}}_{F}$, and ${\mathrm{FM}}_{S}$ under various XCI patterns when β=0.15, F=0 and $\left(q_{f},\;q_{m}\right)=\left(0.3,\;0.3\right),\;\left(0.3,\;0.2\right)$, and $\left(0.2,\;0.3\right),\;$respectively. These figures show that all four subfigures exhibited a similar pattern in power, indicating that the covariates had a very limited impact on the performance of all methods.

In Figure 1, we can see that ${\mathrm{FM}}_{01}$ and ${\mathrm{FM}}_{F}$ were generally less powerful than other methods in all situations. XCMAX4 performed best when γ=0 (XCI-SN) and 2 (XCI-SR). However, when γ=1 (XCI-R), ${\mathrm{FM}}_{02}$ was the most powerful, followed by ${\mathrm{FM}}_{S}$ and XCMAX4. This was expected because ${\mathrm{FM}}_{02}$ is proposed exactly under XCI-R. We also observed that XCMAX4 had a better power than ${\mathrm{FM}}_{S}$ when γ=0.5, while ${\mathrm{FM}}_{S}$ performed slightly better than XCMAX4 when γ=1.5. In both scenarios, ${\mathrm{FM}}_{02}$ was still the most powerful method, but the differences in power between these three methods were generally very small. Notice that the results in Figure 2 and Figure 3 are analogous to those in Figure 1, and thereby the allele frequencies of females and males did not apparently change the power profiles of all of the methods.

Figure 4 plots the powers of XCMAX4, ${\mathrm{FM}}_{01}$, ${\mathrm{FM}}_{02}$, ${\mathrm{FM}}_{F}$, and ${\mathrm{FM}}_{S}$ under the XCI-E pattern with β=0.15. Based on this figure, ${\mathrm{FM}}_{01}$ was uniformly the most powerful in all scenarios as expected, followed by ${\mathrm{FM}}_{S}$ and XCMAX4. ${\mathrm{FM}}_{02}$ was generally less powerful than ${\mathrm{FM}}_{01}$, ${\mathrm{FM}}_{S}$, and XCMAX4, but still performed better than ${\mathrm{FM}}_{F}$. The power results with β=0.15, and F=0.05 are provided in Appendix A (Appendix A), which are similar to those in Figure 1, Figure 2, Figure 3 and Figure 4, indicating that HWD in females had little effect on the power results. The power results with β=0.1116 and 0.1858 are generally consistent with those in Figure 1, Figure 2, Figure 3 and Figure 4, implying that the properties of XCMAX4 did not vary with the magnitude of the genetic effect (see Appendix A in Appendix A). As expected, when the value of β increased, the powers of all methods uniformly increased.

In conclusion, ${\mathrm{FM}}_{01}\;$ and ${\mathrm{FM}}_{02}\;$ can have high power if the underlying XCI pattern is modelled correctly but may be less powerful in other scenarios. In contrast, XCMAX4 retained a relatively good power across a variety of scenarios. Compared to XCMAX4, ${\mathrm{FM}}_{S}$ may suffer from power loss if the SNP is undergoing XCI but will be more powerful under XCI-E. ${\mathrm{FM}}_{F}$ had the overall worst performance and thus is not recommend. It should be noted that, ${\mathrm{FM}}_{01}$, ${\mathrm{FM}}_{02}$, ${\mathrm{FM}}_{F}$, and ${\mathrm{FM}}_{S}$ adopted logistic regression, which is slightly more computationally intensive than XCMAX4 in GWAS because the implementation of the logistic regression requires additional iterations. Compared to the other four methods, testing 2000 SNPs, XCMAX4 saved half the time. The details of time comparisons are given in Appendix A (Appendix A).

## 4. Application to Graves’ Disease Data

Graves’ disease (GD) is an autoimmune disease of hyperthyroidism that is four times more common in women than in men [22,23]. Substantial studies have shown that the genetic background explains about four-fifths of the susceptibility to GD.

Considering the distinct gender bias, it is highly reasonable to speculate that the genes on the X chromosome play an important role in the development of GD. Recently, two independent studies found that rs3827440, a non-synonymous SNP of the GRP174 gene on the X chromosome, was associated with GD. A two-stage GWAS, focused on the Han population in China, first reported this finding, which was further validated in two Caucasian cohorts. There are two alleles at rs3827400, with T being the risk allele and C being the normal one. Table 3 displays the four datasets about rs3827400 mentioned in these two studies. We applied XCMAX4, ${\mathrm{FM}}_{01}$, ${\mathrm{FM}}_{02}$, ${\mathrm{FM}}_{F}$, and ${\mathrm{FM}}_{S}$ to each dataset; the results are shown in Table 4. Note that sex was included as a covariate when calculating the *p* values of XCMAX4, ${\mathrm{FM}}_{01}$, and ${\mathrm{FM}}_{02}$.

This table indicates that none of these methods uniformly performed the best across all four datasets. For the two datasets from the Chinese population, all methods consistently showed that rs3827400 was associated with GD at the $1\times 10^{-4}\;\mathrm{significance}\;\mathrm{level}$. Among these tests, XCMAX4 consistently had the second smallest *p* values. However, the *p* values of all the methods from both Caucasian datasets suggested no such an association at the same significance level probably because of their relatively small sample size. We also observed that XCMAX4 appeared slightly conservative in these scenarios, but this was not surprising because the rhombus formula is less accurate when the *p* value is greater than 0.01.

Because both the Han population and the Caucasian population contained two datasets, we also tested such association at the population level by treating the data source as an additional covariate. The corresponding results are given in Table 5, which are similar to those in Table 4.

## 5. Discussion

This paper proposed a novel robust method, XCMAX4, to test the association between the marker on the X chromosome and a specific disease for case-control design. Our method is an extension of the CMAX3 [24] test on the X chromosome, which can both incorporate the information of XCI and allow for covariates. Unlike the maxLR proposed by Wang et al., XCMAX4 is construted by using the score test, which is more efficient in GWAS because we only need to fit the null model once. Moreover, the maxLR requires permutation to calculate the *p* value, which makes it unappealing in GWAS. In contrast, the *p* value of XCMAX4 can be computed analytically by using the rhombus formula. On the other hand, although ${\mathrm{FM}}_{01}$, ${\mathrm{FM}}_{02}$, ${\mathrm{FM}}_{F}$, and ${\mathrm{FM}}_{S}$ can also adjust for covariates, they do not take various XCI models into consideration and thus may suffer from substantial power loss in some scenarios. However, XCMAX4 can retain a relatively high power by accounting for four special XCI patterns simultaneously. Simulation results showed that XCMAX4 controlled the size well and had robust performance in power. Therefore, we recommend using XCMAX4 for its effectiveness, robustness, and generality. Finally, to help implement XCMAX4 in practice, we provide an R function XCMAX4, which is available at https://github.com/YoupengSU/XCMAX4.git (accessed on 12 April 2022).

## Figures and Tables

**Figure 1 genes-13-00847-f001:**
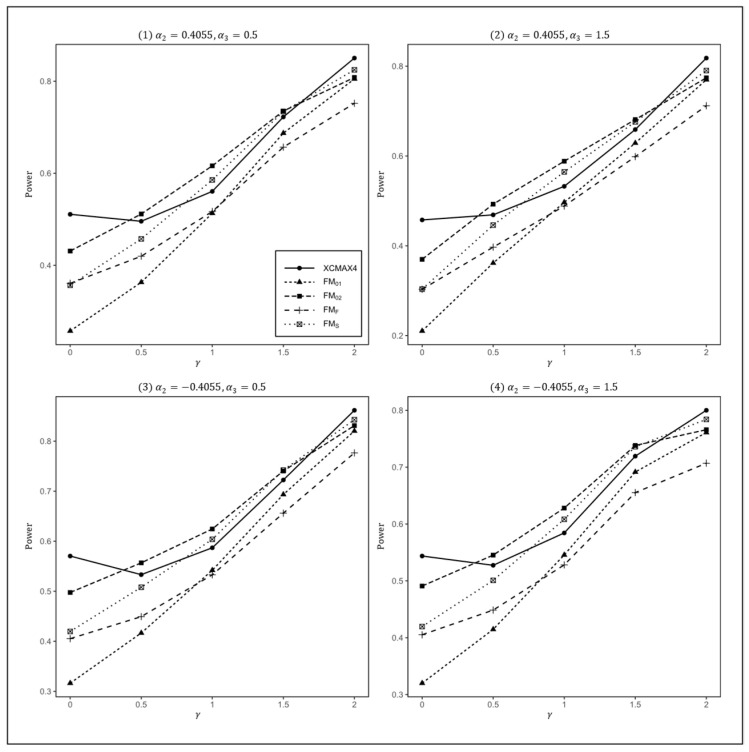
Estimated powers of XCMAX4, ${\mathrm{FM}}_{01}$, ${\mathrm{FM}}_{02}$, ${\mathrm{FM}}_{F}$, and ${\mathrm{FM}}_{S}$ under various XCI models. The simulation was based on 10,000 replicates with β=0.15, $\alpha_{1}=-5$, F=0, and $q_{f}=q_{m}=0.3$.

**Figure 2 genes-13-00847-f002:**
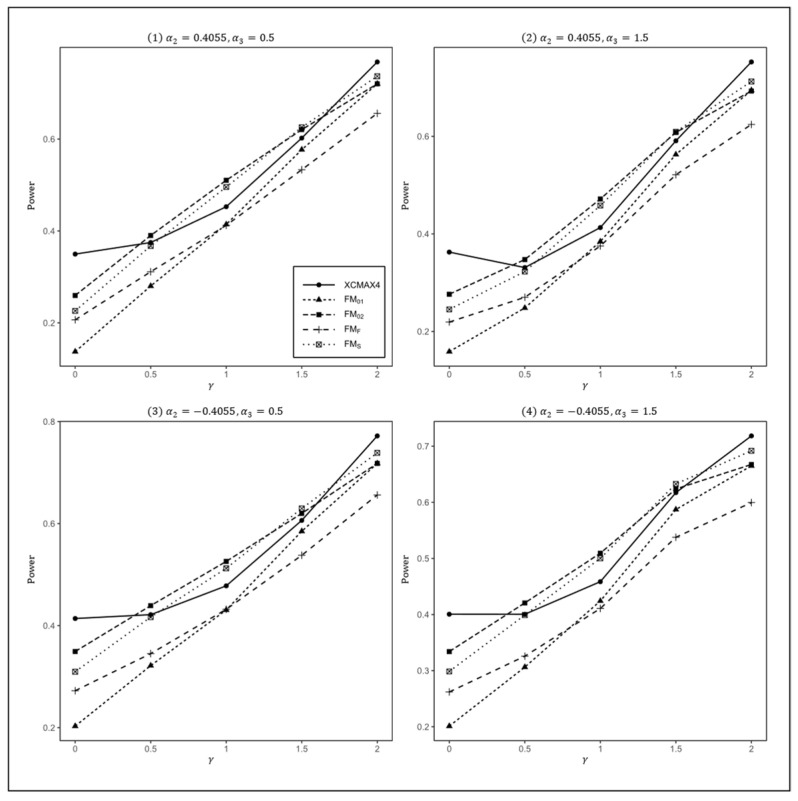
Estimated powers of XCMAX4, ${\mathrm{FM}}_{01}$, ${\mathrm{FM}}_{02}$, ${\mathrm{FM}}_{F}$, and ${\mathrm{FM}}_{S}$ under various XCI models. The simulation was based on 10,000 replicates with β=0.15, $\alpha_{1}=-5$, F=0, $q_{f}=0.3$, and $q_{m}=0.2$.

**Figure 3 genes-13-00847-f003:**
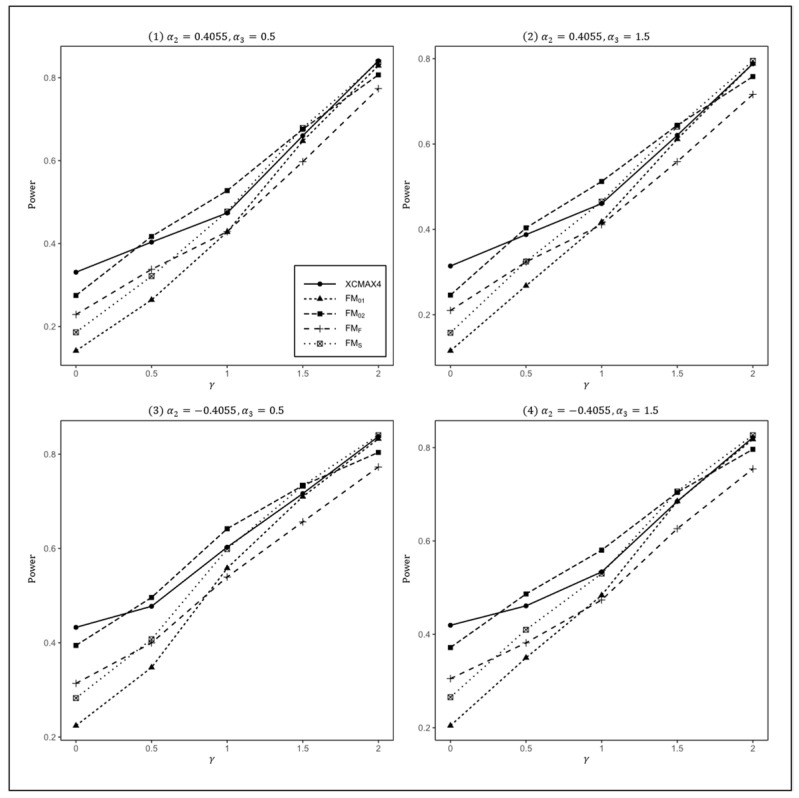
Estimated powers of XCMAX4, ${\mathrm{FM}}_{01}$, ${\mathrm{FM}}_{02}$, ${\mathrm{FM}}_{F}$, and ${\mathrm{FM}}_{S}$ under various XCI models. The simulation was based on 10,000 replicates with β=0.15, $\alpha_{1}=-5$, F=0, $q_{f}=0.2$, and $q_{m}=0.3$.

**Figure 4 genes-13-00847-f004:**
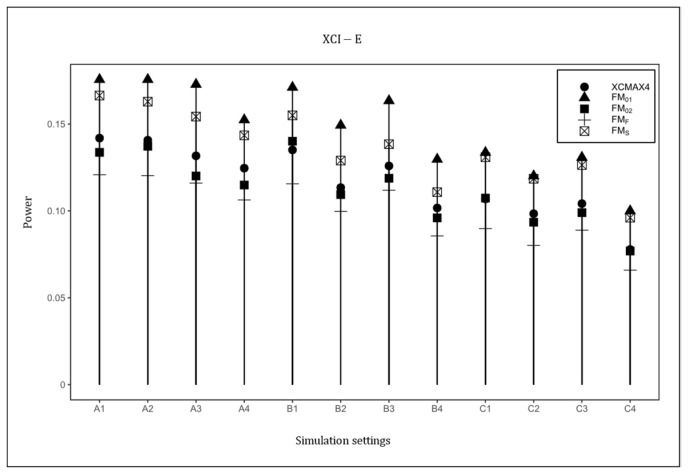
Powers of XCMAX4, ${\mathrm{FM}}_{01}$, ${\mathrm{FM}}_{02}$, ${\mathrm{FM}}_{F}$, and ${\mathrm{FM}}_{S}$ under XCI-E. The simulation was based on 10,000 replicates with β=0.15, $\alpha_{1}=-5$, and F=0. In the horizontal coordinates, “A”, “B”, and “C” represent three combinations of $\left(q_{f},\;q_{m}\right)$: $\left(0.3,0.3\right)$, $\left(0.3,0.2\right)$, and $\left(0.2,0.3\right)$, respectively, and the numbers 1–4 represent four combinations of $\left(\alpha_{2},\alpha_{3}\right)$: $\left(0.4055,0.5\right)$, $\left(0.4055,1.5\right)$, $\left(-0.4055,\;0.5\right)$, and $\left(-0.4055,1.5\right)$, respectively.

**Table 1 genes-13-00847-t001:** The genotypic scores for five genotypes and their corresponding values of $Z_{1}\;$ and $Z_{2}\;$ under the four special XCI patterns.

XCI Pattern	aa	Aa	AA	a	A	$Z_{1}$	$Z_{2}$
XCI-SN	0	0	2	0	2	0	2
XCI-R	0	1	2	0	2	1	2
XCI-SR	0	2	2	0	2	2	2
XCI-E	0	1	2	0	1	1	1

**Table 2 genes-13-00847-t002:** Estimated type I error rate $\left(\times 10^{-4}\right)$ at the nominal significance level $1\times 10^{-4}$ for XCMAX4, ${\mathrm{FM}}_{01}$, ${\mathrm{FM}}_{02}$, ${\mathrm{FM}}_{F}$, and ${\mathrm{FM}}_{S}$ against $q_{f}$, $q_{m}$, $\alpha_{2}$, and $\alpha_{3}$ based on 1,000,000 replicates under HWE.

$q_{f}$	$q_{m}$	$\alpha_{3}\;$	$\alpha_{2}\;=\;0.4005$					$\alpha_{2}\;=\;-0.4005$				
			XCMAX4	$FM_{01}$	$FM_{02}$	$FM_{F}$	$FM_{S}$	XCMAX4	$FM_{01}$	$FM_{02}$	$FM_{F}$	$FM_{S}$
0.1	0.1	0.5	1.03	0.74	0.86	0.95	0.82	0.87	0.93	0.93	0.66	0.98
	0.2		0.88	0.96	0.84	0.86	0.96	1.07	1.02	1.10	0.99	1.02
	0.3		0.91	0.87	0.84	1.10	0.84	0.86	1.02	0.88	0.95	1.00
0.2	0.1		0.94	0.96	0.98	0.93	0.95	0.86	0.85	0.74	0.78	0.79
	0.2		1.02	1.02	0.93	1.12	1.00	1.12	1.43	1.24	1.22	1.39
	0.3		0.90	1.09	0.99	0.76	1.02	1.10	0.99	1.03	0.98	1.01
0.3	0.1		0.87	0.88	0.87	0.88	0.87	0.79	1.01	0.93	0.96	0.91
	0.2		0.93	1.01	1.04	0.77	0.99	0.83	0.94	0.89	0.91	0.93
	0.3		0.83	1.13	0.92	0.95	0.91	0.96	1.15	1.05	1.14	1.06
0.1	0.1	1.5	0.88	1.06	0.93	0.79	1.01	0.88	1.00	0.83	0.77	0.92
	0.2		0.84	0.77	0.82	0.79	0.75	0.92	0.87	0.96	0.98	0.86
	0.3		0.88	1.22	1.08	0.96	1.16	0.99	1.17	1.14	1.07	1.09
0.2	0.1		0.92	0.93	1.06	0.92	1.03	0.81	1.03	0.91	0.96	0.91
	0.2		0.84	1.01	0.92	0.91	0.98	0.86	0.80	0.88	0.85	0.78
	0.3		0.94	1.08	1.14	1.08	1.08	0.85	0.94	0.98	0.98	0.97
0.3	0.1		1.00	1.08	1.03	0.93	1.00	0.99	0.85	1.01	0.96	0.94
	0.2		0.88	1.08	0.91	0.93	0.92	0.93	0.76	0.90	0.83	0.94
	0.3		0.82	0.95	0.86	1.01	0.88	0.98	1.06	0.97	1.10	1.09

**Table 3 genes-13-00847-t003:** Data of rs3827400 related to Graves’ disease in two independent studies.

Dataset	Race	Female Case			Male Case		Female Control			Male Control	
		CC	TC	TT	C	T	CC	TC	TT	C	T
Chu et al. (stage I)	Han	163	508	444	109	232	219	541	367	172	186
Chu et al. (stage II)	Han	471	1606	1298	284	606	584	1344	957	396	526
Szymanski et al. (Warsaw)	Caucasian	146	205	85	53	51	188	229	81	146	104
Szymanski et al. (Gliwice)	Caucasian	58	78	30	20	11	71	73	27	20	10

**Table 4 genes-13-00847-t004:** *p* values of the XCMAX4, ${\mathrm{FM}}_{01}$, ${\mathrm{FM}}_{02}$, ${\mathrm{FM}}_{F}$, and ${\mathrm{FM}}_{S}$ tests from four datasets.

Dataset	XCMAX4	$FM_{01}$	$FM_{02}$	$FM_{F}$	$FM_{S}$
Chu et al. (stage I) $\left(\times 10^{-8}\right)$	1.573	9.513	0.507	1.832	1.731
Chu et al. (stage II) $\left(\times 10^{-15}\right)$	0.847	7.764	0.561	4.108	1.144
Szymanski et al. (Warsaw) $\left(\times 10^{-1}\right)$	1.083	0.491	0.395	1.038	0.410
Szymanski et al. (Gliwice) $\left(\times 10^{-1}\right)$	5.967	2.515	2.800	5.500	2.628

**Table 5 genes-13-00847-t005:** *p* values of XCMAX4, ${\mathrm{FM}}_{01}$, ${\mathrm{FM}}_{02}$, ${\mathrm{FM}}_{F}$, and ${\mathrm{FM}}_{S}$ tests from Han and Caucasian populations.

Population	XCMAX4	$FM_{01}$	$FM_{02}$	$FM_{F}$	$FM_{S}$
Han $\left(\times 10^{-22}\right)$	1.275	55.347	0.285	1.792	1.444
Caucasian $\left(\times 10^{-2}\right)$	6.932	2.571	2.553	5.795	1.993

## Data Availability

The real data used in this study are available from two published papers at https://dx.doi.org/10.1136%2Fjmedgenet-2013-101595, and https://doi.org/10.1111/tan.12259 (assessed on 7 April 2022).

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
