# Peer review of "XCMAX4: A Robust X Chromosomal Genetic Association Test Accounting for Covariates"

_genes, 2022, doi:10.3390/genes13050847_

Round 1

Reviewer 1 Report

The unified robust test for X chromosomal genetic variants presented in this paper is interesting and quite useful. I just have a few comments that can improve and strengthen the paper.

  1. Providing benchmarking results for computational resource requirements for the proposed method, and comparing with a few existing methods can be useful to evaluate its practical applicability.
  2. Page 5, model (2), why not use all four types of G scores in the model? Why only use two G scores? If my intuition is correct, then using all four types of G scores allows one to calculate all the marginal correlations from one model only.

Reviewer 2 Report

This is an interesting paper on a novel association test for SNPs on X-chromosome. I have some questions:

  1. The size of the test is only evaluated at one significance level (10^{-4}). It would be more convincing to present the sizes at multiple significance levels.
  2. Table 2 is confusing. It presents the type I error rate, but it is not clear what the denominator is. For example, does 1.03 means 1.03 errors per 100 tests or 10,000 tests?
  3. When evaluating power, the authors uses only one beta value (0.15). It would be helpful to use multiple beta values to see how the power changes with beta.

Two lines above Table 2, "litter impact" should be "little impact".

Round 2

Reviewer 2 Report

This is a revision. The authors have addressed my previous concerns sufficiently.